# Deep Semantic Role Labeling: What Works and What's Next

## Abstract

We introduce a new deep learning model for semantic role labeling (SRL) that significantly improves the state of the art, along with detailed analyses to reveal its strengths and limitations. We use a deep highway BiLSTM architecture with constrained decoding, while observing a number of recent best practices for initialization and regularization. Our 8-layer ensemble model achieves 83.2 F1 on the CoNLL 2005 test set and 83.4 F1 on CoNLL 2012, roughly a 10% relative error reduction over the previous state of the art. Extensive empirical analysis of these gains show that (1) deep models excel at recovering long-distance dependencies but can still make surprisingly obvious errors, and (2) that there is still room for syntactic parsers to improve these results. All code and models will be publicly released.

## 1 Introduction

Semantic role labeling (SRL) systems aim to recover the predicate-argument structure of a sentence, to determine essentially "who did what to whom", "when", and "where." Recently breakthroughs involving end-to-end deep models for SRL without syntactic input (Zhou and Xu, 2015; Marcheggiani et al., 2017) seem to overturn the long-held belief that syntactic parsing is a prerequisite for this task (Punyakanok et al., 2008). In this paper, we show that this result can be pushed further using deep highway bidirectional LSTMs with constrained decoding, again significantly moving the state of the art (another 2 points on CoNLL 2005). We also present careful empirical analysis to determine what works well and what might be done to progress even further.

Our model combines a number of best practices in the recent deep learning literature. Following Zhou and Xu (2015), we treat SRL as a BIO tagging problem and use deep bidirectional LSTMs. However, we differ by (1) simplifying the input and output layers, (2) introducing highway connections (Srivastava et al., 2015; Zhang et al., 2016), (3) using recurrent dropout (Gal, 2015), (4) decoding with BIO-constraints, and (5) ensembling with a product of experts. Our model gives a 10% relative error reduction over previous state of the art on the test sets of CoNLL 2005 and 2012. All code and models will be publicly released.

We present a set of detailed error analyses to better understand the performance gains, including (1) design choices on architecture, initialization, and regularization that have a surprisingly large impact on model performance, (2) analyses of different types of prediction errors, e.g. showing that deep models excel at predicting long-distance dependencies but still struggles with known challenges such as PP-attachment errors and adjunct-argument distinctions, (3) a careful analysis of the role of syntax, showing that there is significant room for improvement given oracle syntax but errors from existing parsers prevent effective use in SRL.

In summary, our main contributions are:

- A new state-of-the-art deep network for end-to-end SRL, supported by code and models that will be publicly available.
- An in-depth error analysis indicating where the models work well and where they still struggle, including discussion of structural consistency and long-distance dependencies.
- Experiments that point toward directions for future improvements, including a detailed discussion of how and when syntactic parsers could be used to improve these results.

## 2 Model

Two major factors contribute to the success of our deep SRL model: (1) applying recent advances in training deep recurrent neural networks such as highway connections (Srivastava et al., 2015; Zhang et al., 2016) and RNN-dropouts (Gal, 2015)[1]; and (2) using an A* decoding algorithm, similar to Lewis and Steedman (2014) to enforce structural consistency at prediction time, without adding more complexity to the training process.

Formally, the goal of our task is to predict a sequence $\boldsymbol{y}$ given a sentence-predicate pair $(\boldsymbol{w}, v)$ as input. Each $y_i$ belongs to a discrete set of BIO tags $\boldsymbol{T}$. Words outside argument spans have the tag O, and words at the beginning and inside of argument with role $r$ have the tags $B_r$ and $I_r$ respectively. Let $n = |\boldsymbol{w}| = |\boldsymbol{y}|$ be the length of the sequence.

Predicting an SRL structure under our model involves finding the highest-scoring tag sequence over the space of all possibilities $\mathcal{Y}$:

$$\hat{\boldsymbol{y}} = \operatorname*{argmax}_{\boldsymbol{y} \in \mathcal{Y}} f(\boldsymbol{w}, \boldsymbol{y}) \qquad (1)$$

We use a deep bidirectional LSTM (BiLSTM) to learn a locally decomposed scoring function conditioned on the input: $\sum_{t=1}^{n} \log p(y_t \mid \boldsymbol{w})$.

To incorporate additional information (e.g. structural consistency, syntactic input), we augment the scoring function with penalization terms:

$$f(\boldsymbol{w}, \boldsymbol{y}) = \sum_{t=1}^{n} \log p(y_t \mid \boldsymbol{w}) - \sum_{c \in \mathcal{C}} c(\boldsymbol{w}, y_{1:t}) \quad (2)$$

Each constraint function $c$ applies a *non-negative* penalty given the input $\boldsymbol{w}$ and a length-$t$ prefix $y_{1:t}$. These constraints can be hard or soft depending on whether the penalties are finite.

### 2.1 Deep BiLSTM Model

Our model computes the distribution over tags using stacked BiLSTMs, which we define as follows:

$$\boldsymbol{i}_{l,t} = \sigma(\mathbf{W}_{i}^{l}[\boldsymbol{h}_{l,t+\delta_l}, \boldsymbol{x}_{l,t}] + \boldsymbol{b}_{i}^{l}) \qquad (3)$$

$$\boldsymbol{o}_{l,t} = \sigma(\mathbf{W}_{o}^{l}[\boldsymbol{h}_{l,t+\delta_l}, \boldsymbol{x}_{l,t}] + \boldsymbol{b}_{o}^{l}) \qquad (4)$$

$$\boldsymbol{f}_{l,t} = \sigma(\mathbf{W}_{f}^{l}[\boldsymbol{h}_{l,t+\delta_l}, \boldsymbol{x}_{l,t}] + \boldsymbol{b}_{f}^{l} + 1) \qquad (5)$$

$$\tilde{\boldsymbol{c}}_{l,t} = \tanh(\mathbf{W}_{c}^{l}[\boldsymbol{h}_{l,t+\delta_l}, \boldsymbol{x}_{l,t}] + \boldsymbol{b}_{c}^{l}) \qquad (6)$$

$$\boldsymbol{c}_{l,t} = \boldsymbol{i}_{l,t} \circ \tilde{\boldsymbol{c}}_{l,t} + \boldsymbol{f}_{l,t} \circ \boldsymbol{c}_{t+\delta_l} \qquad (7)$$

$$\boldsymbol{h}_{l,t} = \boldsymbol{o}_{l,t} \circ \tanh(\boldsymbol{c}_{l,t}) \qquad (8)$$

---

[1]We thank Mingxuan Wang for suggesting highway connections with simplified inputs and outputs. Part of our model is extended from his unpublished implementation.

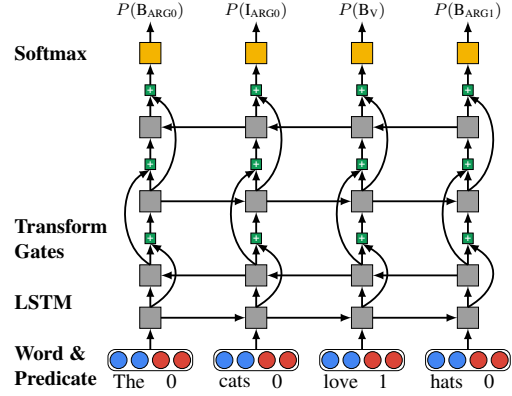

Figure 1: Highway LSTM with four layers. The curved connections represent highway connections, and the plus symbols represent transform gates that control the inter-layer information flow.

where $x_{l,t}$ is the input to the LSTM at layer $l$ and timestep $t$. $\delta_l$ is either 1 or $-1$, indicating the directionality of the LSTM at layer $l$.

To stack the LSTMs in an interleaving pattern, as proposed by Zhou and Xu (2015), the layer-specific inputs $x_{l,t}$ and directionality $\delta_l$ are arranged in the following manner:

$$\boldsymbol{x}_{l,t} = \begin{cases} [\mathbf{W}_{\text{emb}}(w_t), \mathbf{W}_{\text{mask}}(t = v)] & l = 1 \\ \boldsymbol{h}_{l-1,t} & l > 1 \end{cases} \quad (9)$$

$$\delta_l = \begin{cases} 1 & \text{if } l \text{ is even} \\ -1 & \text{otherwise} \end{cases} \qquad (10)$$

The input vector $\boldsymbol{x}_{1,t}$ is the concatenation of token $w_t$'s word embedding and an embedding of the binary feature $(t = v)$ indicating whether the current word is the given predicate.

Finally, the locally normalized distribution over output tags is computed via a softmax layer:

$$p(y_t \mid \boldsymbol{x}) \propto \exp(\mathbf{W}_{\text{tag}}^{y} \boldsymbol{h}_{L,t} + \boldsymbol{b}_{\text{tag}}) \qquad (11)$$

**Highway Connections** To alleviate the vanishing gradient problem when training deep BiLSTMs, we use gated highway connections described in Zhang et al. (2016); Srivastava et al. (2015). We include *transform gates* $\boldsymbol{r}_t$ to control the weight of linear and non-linear transformations between layers (See Figure 1). The output $\boldsymbol{h}_{l,t}$ is changed to:

$$\boldsymbol{r}_{l,t} = \sigma(\mathbf{W}_{r}^{l}[\boldsymbol{h}_{l,t-1}, \boldsymbol{x}_t] + \boldsymbol{b}_{r}^{l}) \qquad (12)$$

$$\boldsymbol{h}_{l,t}' = \boldsymbol{o}_{l,t} \circ \tanh(\boldsymbol{c}_{l,t}) \qquad (13)$$

$$\boldsymbol{h}_{l,t} = \boldsymbol{r}_{l,t} \circ \boldsymbol{h}_{l,t}' + (1 - \boldsymbol{r}_{l,t}) \circ \mathbf{W}_{h}^{l} \boldsymbol{x}_{l,t} \qquad (14)$$

**Recurrent Dropout** To reduce over-fitting, we use dropout as described in Gal (2015). A shared dropout mask $\boldsymbol{z}_l$ is applied to the hidden state:

$$\widetilde{\boldsymbol{h}}_{l,t} = \boldsymbol{r}_{l,t} \circ \boldsymbol{h}'_{l,t} + (1 - \boldsymbol{r}_{l,t}) \circ \mathbf{W}_{\mathrm{h}}^{l} \boldsymbol{x}_{l,t} \quad (15)$$

$$\boldsymbol{h}_{l,t} = \boldsymbol{z}_l \circ \widetilde{\boldsymbol{h}}_{l,t} \quad (16)$$

$\boldsymbol{z}_l$ is shared across timesteps at layer $l$ to avoid amplifying the dropout noise along the sequence.

## 2.2 Constrained A* Decoding

The approach described so far does not model any dependencies between the output tags. To incorporate constraints on the output structure at decoding time, we use A* search over tag prefixes for decoding. Starting with an empty sequence, the tag sequence is built from left to right. The score for a partial sequence with length $t$ is defined as:

$$f(\boldsymbol{w}, y_{1:t}) = \sum_{i=1}^{t} \log p(y_i \mid \boldsymbol{w}) - \sum_{c \in \mathcal{C}} c(\boldsymbol{w}, y_{1:i}) \quad (17)$$

An admissible A* heuristic can be computed efficiently by summing over the best possible tags for all timesteps after $t$:

$$g(\boldsymbol{w}, y_{1:t}) = \sum_{i=t+1}^{n} \max_{y_i \in \boldsymbol{T}} \log p(y_i \mid \boldsymbol{w}) \quad (18)$$

Exploration of the prefixes is determined by an agenda $\mathcal{A}$ which is sorted by $f(\boldsymbol{w}, y_{1:t}) + g(\boldsymbol{w}, y_{1:t})$. In the worst case, A* explores exponentially many prefixes, but because the distribution $p(y_t \mid \boldsymbol{w})$ learned by the BiLSTM models is very peaked, the algorithm is efficient in practice. We list some example constraints as follows:

**BIO Constraints** These constraints reject any sequence that does not produce valid BIO transitions, such as $B_{\mathrm{ARG0}}$ followed by $I_{\mathrm{ARG1}}$.

**SRL Constraints** Punyakanok et al. (2008); Täckström et al. (2015) described a list of SRL-specific global constraints:

- Unique core roles (U): Each core role (ARG0-ARG5, ARGA) should appear at most once for each predicate.
- Continuation roles (C): A continuation role C-X can exist only when its base role X is realized before it.
- Reference roles (R): A reference role R-X can exist only when its base role X is realized (not necessarily before R-X).

We only enforce U and C constraints, since the R constraints are more commonly violated in gold data, and enforcing them results in worse performance (see discussions in Section 4.3).

**Syntactic Constraints** We can enforce consistency with a given parse tree by rejecting or penalizing arguments that are not constituents. In Section 4.4, we will discuss the motivation behind using syntactic constraints and experimental results using both predicted and gold syntax.

## 3 Experiments

### 3.1 Datasets

We measure the performance of our SRL system on two PropBank-style, span-based SRL datasets: CoNLL-2005 (Carreras and Màrquez, 2005) and CoNLL-2012 (Pradhan et al., 2013)[2]. Both datasets provide gold predicates (their index in the sentence) as part of the input. Therefore, each provided predicate corresponds to one training/test tag sequence. We follow the train-development-test split for both datasets and use the official evaluation script from the CoNLL 2005 shared task for evaluation on both datasets.

### 3.2 Model Setup

Our network consists of 8 BiLSTM layers (4 forward LSTMs and 4 reversed LSTMs) with 300-dimensional hidden units, and a softmax layer for predicting the output distribution.

**Initialization** All the weight matrices in BiLSTMs are initialized with random orthonormal matrices as described in Saxe et al. (2013).

All the tokens are lower-cased and initialized with 100-dimensional GloVe embeddings pre-trained on 6B tokens (Pennington et al., 2014) and updated during training. Tokens that are not covered by GloVe are replaced with a randomly initialized UNK embedding.

**Training** We use Adadelta (Zeiler, 2012) with $\epsilon = 1e^{-6}$ and $\rho = 0.95$ and mini-batches of size 80. We set RNN-dropout probability to 0.1 and clip gradients with norm larger than 1. All the models are trained for 500 epochs with early stopping based on development results [3].

---

[2]We used the version of OntoNotes downloaded at: http://cemantix.org/data/ontonotes.html.

[3]Training the full model on CoNLL 2005 takes about 5 days on a single Titan X Pascal GPU.

| Method | Development | | | | WSJ Test | | | | Brown Test | | | | Combined |
|---|---|---|---|---|---|---|---|---|---|---|---|---|---|
| | P | R | F1 | Comp. | P | R | F1 | Comp. | P | R | F1 | Comp. | F1 |
| Ours (PoE) | **83.1** | **82.4** | **82.7** | **64.1** | **85.0** | **84.3** | **84.6** | **66.5** | **74.9** | **72.4** | **73.6** | **46.5** | **83.2** |
| Ours | 81.6 | 81.6 | 81.6 | 62.3 | 83.1 | 83.0 | 83.1 | 64.3 | 72.9 | 71.4 | 72.1 | 44.8 | 81.6 |
| Zhou | - | - | 79.6 | - | - | - | 82.8 | - | - | - | 69.4 | - | 81.1 |
| FitzGerald (Struct.,PoE) | 81.2 | 76.7 | 79.2 | 55.2 | 82.5 | 78.2 | 80.3 | 57.3 | 74.5 | 70.0 | 72.2 | 41.3 | - |
| Täckström (Struct.) | 81.2 | 76.2 | 78.6 | 54.4 | 82.3 | 77.6 | 79.9 | 56.0 | 74.3 | 68.6 | 71.3 | 39.8 | - |
| Toutanova (Ensemble) | - | - | 78.6 | 58.7 | 81.9 | 78.8 | 80.3 | 60.1 | - | - | 68.8 | 40.8 | - |
| Punyakanok (Ensemble) | 80.1 | 74.8 | 77.4 | 50.7 | 82.3 | 76.8 | 79.4 | 53.8 | 73.4 | 62.9 | 67.8 | 32.3 | 77.9 |

Table 1: Experimental results on CoNLL 2005, in terms of precision (P), recall (R), F1 and percentage of completely correct predicates (Comp.). We report results of our best single and ensemble (PoE) model. The comparison models are Zhou and Xu (2015), FitzGerald et al. (2015), Täckström et al. (2015), Toutanova et al. (2008) and Punyakanok et al. (2008).

| | Development | | | | Test | | | |
|---|---|---|---|---|---|---|---|---|
| | P | R | F1 | Comp. | P | R | F1 | Comp. |
| Ours (PoE) | **83.5** | **83.2** | **83.4** | **67.5** | **83.5** | **83.3** | **83.4** | **68.5** |
| Ours | 81.7 | 81.4 | 81.5 | 64.6 | 81.8 | 81.6 | 81.7 | 66.0 |
| Zhou | - | - | 81.1 | - | - | - | 81.3 | - |
| FitzGerald (Struct.,PoE) | 81.0 | 78.5 | 79.7 | 60.9 | 81.2 | 79.0 | 80.2 | 62.6 |
| Täckström (Struct.) | 80.5 | 77.8 | 79.1 | 60.1 | 80.6 | 78.2 | 79.4 | 61.8 |
| Pradhan (revised) | 78.5 | 76.6 | 77.5 | 55.8 | 78.5 | 76.6 | 77.5 | 55.8 |

Table 2: Experimental results on CoNLL 2012 in the same metrics as above. We compare our best single and ensemble (PoE) models against Zhou and Xu (2015), FitzGerald et al. (2015), Täckström et al. (2015) and Pradhan et al. (2013).

**Ensembling** We use a product of experts (Hinton, 2002) to combine predictions of 5 models, each trained on 80% of the training corpus and validated on the remaining 20%. For the CoNLL 2012 corpus, we split the training data from each sub-genre into 5 folds, such that the training data will have similar genre distributions.

**Constrained Decoding** We experimented with different types of constraints on the CoNLL 2005 and CoNLL 2012 development sets. Only the BIO hard constraints significantly improve over the ensemble model. Therefore, in our final results, we only use BIO hard constraints during decoding [4].

### 3.3 Results

In Table 1 and 2, we compare our best single and ensemble model with previous work. Our ensemble (PoE) has an absolute improvement of 2.1 F1 on CoNLL 2005 and 2.3 on CoNLL 2012 over the previous state of the art. Our single model also achieves more than 0.4 improvement on both datasets. In comparison with the best reported results, our percentage of completely correct predicates improves by 5.9%. While the continuing trend of improving SRL without syntax seems to

---

[4] A* search in this setting finds the optimal sequence for all sentences and is therefore equivalent to Viterbi decoding.

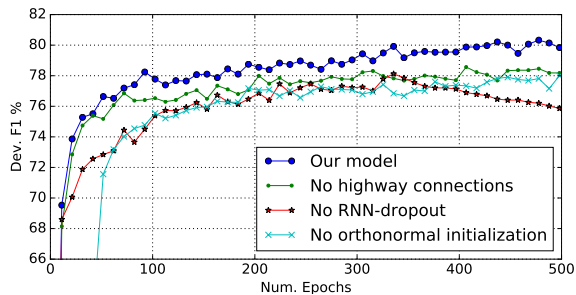

Figure 2: Learning curve of various ablations. The combination of highway layers, orthonormal parameter initialization and recurrent dropout is crucial to achieving strong performance. The numbers shown here are without constrained decoding.

suggest that neural end-to-end systems no longer needs parsers, our analysis in Section 4.4 will show that accurate syntactic information can improve these deep models.

### 3.4 Ablations

Figure 2 shows learning curves of our model ablations on the CoNLL 2005 development set. We ablate our full model by removing highway connections, RNN-dropout and orthonormal initialization independently. Without dropout, the model overfits at around 300 epochs at 78% F1. Or-

thonormal parameter initialization is surprisingly important—without this, the model achieves only 65% F1 within the first 50 epochs. All 8 layer ablations suffer a loss of more than 1.7% in absolute F1 compared to the full model.

## 4 Analysis

To better understand our deep SRL model and its relation to previous work, we address the following questions with a suite of empirical analyses:

- What is the model good at and what kind of mistakes does the model make?
- How well do LSTMs model global structural consistency, despite conditionally independent tagging decisions?
- Is our model implicitly learning syntax, and could explicitly modeling syntax still help?

All the analysis in this section is done on the CoNLL 2005 development set unless otherwise stated. We are also able to compare to previous systems with model predictions available online (Punyakanok et al., 2005; Pradhan et al., 2005)[5].

### 4.1 Error Types Breakdown

Inspired Kummerfeld et al. (2012), we define a set of oracle transformations that fix various prediction errors and observe the relative improvement after each operation (See Table 3). Figure 3 shows how our work compare to the previous systems in terms of different types of mistakes. The analysis shows that while our model makes a similar number of labeling errors to traditional syntax-based systems, it has far fewer missing arguments (perhaps due to parser errors making some arguments difficult to recover for syntax-based systems).

**Label Confusion** As shown in Table 3, our system most commonly makes labeling errors, where the predicted span is an argument but the role was incorrectly labeled. Table 4 shows a confusion matrix for the most frequent labels. The model often confuses ARG2 with AM-DIR, AM-LOC and AM-MNR. These confusions can arise due to the use of ARG2 in many verb frames to represent semantic relations such as direction or location. For example, ARG2 in the frame *move.01* is defined as *Arg2-GOL: destination* [6]. This type of argument-

---

[5]Model predictions of CoNLL 2005 systems: http://www.cs.upc.edu/~srlconll/st05/st05.html
[6]Source: Unified verb index: http://verbs.colorado.edu.

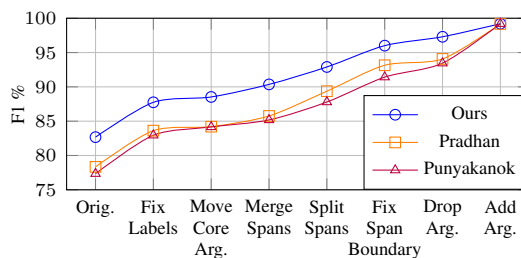

Figure 3: Performance after doing each type of oracle transformation, compared to two strong non-neural baselines. The gap is closed after the *Add Arg.* transformation, showing how our approach is gaining from predicting more arguments than traditional systems.

| Operation | Description | % |
|---|---|---|
| Fix Labels | Correct the span label its boundary matches gold. | 29.3 |
| Move Arg. | Move an unique core argument to its correct position. | 4.5 |
| Merge Spans | Combine two predicted spans into a gold span if they are separated by at most one word. | 10.6 |
| Split Spans | Split a predicted span into two gold spans that are separated by at most one word. | 14.7 |
| Fix Boundary | Correct the boundary of a span if its label matches an overlapping gold span. | 18.0 |
| Drop Arg. | Drop a predicted argument that does not overlap with any gold span. | 7.4 |
| Add Arg. | Add a gold argument does not overlap with any predicted span. | 11.0 |

Table 3: Oracle transformations paired with the relative error reduction after each operation. All the operations are permitted only if they do not cause any overlapping arguments.

adjunct distinction is known to be difficult (Kingsbury et al., 2002), and it is not surprising that our neural model has many such failure cases.

**Attachment Mistakes** A second common source of error is reflected by the *Merge Spans* transformation (10.6%) and the *Split Spans* transformation (14.7%). These errors are closely tied to prepositional phrase (PP) attachment errors, which are also known to be some of the biggest challenges for linguistic analysis (Kummerfeld et al., 2012). Figure 4 shows the distribution of syntactic span labels involved in an attachment mistake, where 62% of the syntactic spans are prepositional phrases. For example, in *Sumitomo financed the acquisition from Sears* our model mistakenly labels the prepositional phrase *from*

| pred. \ gold | A0 | A1 | A2 | A3 | ADV | DIR | LOC | MNR | PNC | TMP |
|---|---|---|---|---|---|---|---|---|---|---|
| A0 | 76 | 13 | 6 | 14 | 2 | 0 | 0 | 0 | 0 | 0 |
| A1 | 16 | 74 | 25 | 0 | 0 | 18 | 9 | 11 | 19 | 2 |
| A2 | 2 | 5 | 31 | 52 | 10 | 45 | 26 | 46 | 19 | 0 |
| A3 | 1 | 0 | 1 | 57 | 2 | 0 | 0 | 0 | 19 | 2 |
| ADV | 0 | 0 | 0 | 5 | 33 | 0 | 11 | 33 | 19 | 5 |
| DIR | 0 | 0 | 3 | 5 | 0 | 27 | 9 | 2 | 0 | 0 |
| LOC | 1 | 2 | 7 | 0 | 2 | 0 | 34 | 11 | 0 | 2 |
| MNR | 1 | 0 | 7 | 29 | 21 | 0 | 0 | 43 | 0 | 3 |
| PNC | 0 | 1 | 3 | 5 | 0 | 9 | 3 | 2 | 44 | 0 |
| TMP | 0 | 2 | 3 | 0 | 26 | 9 | 20 | 7 | 0 | 71 |

Table 4: Confusion matrix of span labels, showing the percentage of predicted labels for each gold label. We only count predicted arguments that match gold span boundaries.

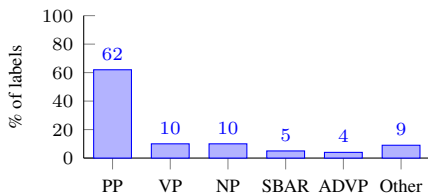

Figure 4: For cases where our model either splits a gold span into two ($Z \rightarrow XY$) or merges two gold constituents ($XY \rightarrow Z$), we show the distribution of syntactic labels for the $Y$ span. Results show the major cause of these errors is inaccurate prepositional phrase attachment.

*Sears* as the ARG2 of *financed*, whereas it should instead attach to *acquisition*.

### 4.2 Long-range Dependencies

To analyze the model's ability to capture long-range dependencies, we compute the F1 of our model on arguments with various distances to the predicate. Figure 5 shows that for all models, performance tends to degrade for arguments further from the predicate. Interestingly, the gap between shallow and deep models becomes much larger for the long-distance predicate-argument structures. The absolute gap between our 2 layer and 8 layer models is 3-4 F1 for arguments that are within 2 words to the predicate, and 5-6 F1 for arguments that are farther away from the predicate. Surprisingly, the neural model performance deteriorates less severely on long-range dependencies than traditional syntax-based models.

### 4.3 Structural Consistency

We can quantify two types of structural consistencies: the BIO constraints and the SRL-specific constraints. Via our ablation study, we show that deeper BiLSTMs are better at enforcing these structural consistencies, although not perfectly.

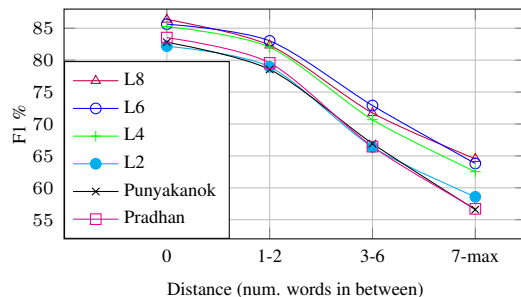

Figure 5: F1 by surface distance between predicates and arguments. Performance degrades least rapidly on long-range arguments for the deeper neural models.

**BIO Violations** The BIO format requires argument spans to begin with a B tag. Any I tag directly following an O tag or a tag with different label is considered a violation. Table 5 shows the number of BIO violations per token for BiLSTMs with different depths. The number of BIO violations decreases when we use a deeper model. The gap is biggest between 2-layer and 4-layer models, and diminishes after that.

It is surprising that although the deeper models generate impressively accurate token-level predictions, they still make enough BIO errors to significantly hurt performance—when these constraints are simple enough to be enforced by trivial rules. To understand the reason behind it, we compare the average entropy between tokens involved BIO violations with the averaged entropy of all tokens. For the 8-layer model, the average entropy on these tokens is 30 times higher than the averaged entropy on all tokens. This suggests that BIO inconsistencies occur when there is some ambiguity. For example, if the model is unsure whether two consecutive words should belong to an ARG0 or ARG1, it might generate inconsistent BIO sequences such as $B_{ARG0}, I_{ARG1}$ when decoding at the token level. Using BIO-constrained decoding can resolve this ambiguity and result in a structurally consistent solution.

**SRL Structure Violations** The model predictions can also violate the SRL-specific constraints commonly used in prior work (Punyakanok et al., 2008; Täckström et al., 2015). As shown in Table 6, the model occasionally violates these SRL constraints. With our constrained decoding algorithm, it is straightforward to enforce the unique core roles (U) and continuation roles (C) constraints during decoding. The constrained de-

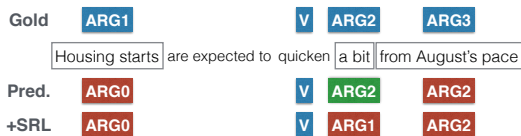

Figure 6: Example where performance is hurt by enforcing the constraint that core roles may only occur once (+SRL).

coding results are shown with the model named *L8+PoE+SRL* in Table 6.

Although the violations are eliminated, the performance does not significantly improve. This is mainly due to two factors: (1) the model often already satisfies these constraints on its own, so the number of violations to be fixed are relatively small, and (2) the gold SRL structure sometimes violate the constraints and enforcing hard constraints can hurt performance. Figure 6 shows a sentence in the CoNLL 2005 development set. Our original model produces two ARG2s for the predicate *quicken*, and this violates the SRL constraints. When the $A^*$ decoder fixes this violation, it changes the first ARG1 into ARG2 because ARG0, ARG1, ARG2 is a more frequent pattern in the training data and has higher overall score.

### 4.4 Can Syntax Still Help SRL?

The Propbank-style SRL formalism is closely tied to syntax (Bonial et al., 2010; Weischedel et al., 2011). In Table 6, we show that 98.7% of the gold SRL arguments match an unlabeled constituent in the gold syntax tree. Similar to some recent work (Zhou and Xu, 2015), our model achieved strong performance without directly modeling syntax. A natural question follows: are neural SRL models implicitly learning syntax? Table 6 shows the trend of deeper models making predictions that are more consistent with the gold syntax in terms of span boundaries. With our best model, 94.3% of the predicted arguments spans are part of the gold parse tree. This consistency is on par with previous CoNLL 2005 systems that directly model constituency and use predicted parse trees as features (Punyakanok, 95.3% and Pradhan, 93.0%).

**Constrained Decoding with Syntax** The above analysis raises a further question: would improving consistency with syntax provide improvements for SRL? Our constrained decoding algorithm described in Section 2.2 enables us to inject syntax as a decoding constraint without having to re-train

| Model (no BIO) | Accuracy | | Violations | Avg. Entropy | |
|---|---|---|---|---|---|
| | F1 | Token | BIO | All | BIO |
| L8+PoE | 81.5 | 91.5 | 0.07 | 0.02 | 0.72 |
| L8 | 80.5 | 90.9 | 0.07 | 0.02 | 0.73 |
| L6 | 80.1 | 90.3 | 0.06 | 0.02 | 0.72 |
| L4 | 79.1 | 90.2 | 0.08 | 0.02 | 0.70 |
| L2 | 74.6 | 88.4 | 0.18 | 0.03 | 0.66 |

Table 5: Comparison of BiLSTM models without BIO decoding. We compare F1 and token-level accuracy (Token), averaged BIO violations per token (BIO), overall model entropy (All) model entropy at tokens involved in BIO violations (BIO). Increasing the depth of the model beyond 4 does not produce more structurally consistent output, emphasizing the need for constrained decoding.

| Model or Oracle | F1 | Syn % | SRL-Violations | | |
|---|---|---|---|---|---|
| | | | U | C | R |
| Gold | 100.0 | 98.7 | 24 | 0 | 61 |
| L8+PoE | 82.7 | 94.3 | 37 | 3 | 68 |
| L8 | 81.6 | 94.0 | 48 | 4 | 73 |
| L6 | 81.4 | 93.7 | 39 | 3 | 85 |
| L4 | 80.5 | 93.2 | 51 | 3 | 84 |
| L2 | 77.2 | 91.3 | 96 | 5 | 72 |
| L8+PoE+SRL | 82.8 | 94.2 | 5 | 1 | 68 |
| L8+PoE+AutoSyntax | 83.2 | 96.1 | 113 | 3 | 68 |
| L8+PoE+GoldSyntax | 85.0 | 97.6 | 102 | 3 | 68 |
| Punyakanok | 77.4 | 95.3 | 0 | 0 | 0 |
| Pradhan | 78.3 | 93.0 | 84 | 3 | 58 |

Table 6: Comparison of models with different depths and decoding constraints (in addition to BIO) as well as two previous systems. We compare F1, unlabeled agreement with gold constituency (Syn%) and each type of SRL-constraint violations (**U**nique core roles, **C**ontinuation roles and **R**eference roles). Our best model produces a similar number of constraint violations to the gold annotation, explaining why deterministically enforcing these constraints is not helpful.

the model. In Figure 7, we compare the SRL accuracy with syntactic constraints specified by gold parse or automatic parses. The penalty of disagreeing with the parse tree is a single parameter dictating how much the model should trust the provided syntax. When using gold syntax, the predictions improve up to 2 F1 as the penalty increases. The improvement from using Choe's parser (Choe and Charniak, 2016) is much smaller, while using the Charniak parser (Charniak, 2000) hurts performance if the model places too much trust in it. These results suggest that high-quality syntax can still make a large impact on SRL.

A known challenge for syntactic parsers is robustness on out-of-domain data. Therefore we

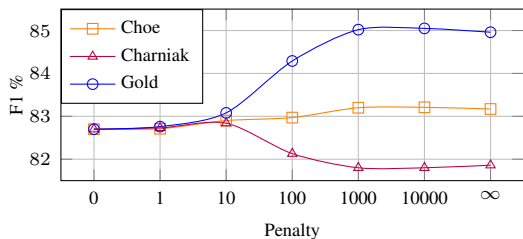

Figure 7: Performance of syntax-constrained decoding as the non-constituent penalty increases for syntax from two parsers (from Choe and Charniak (2016) and Charniak (2000)) and gold syntax. The best existing parser gives a small improvement, but the improvement from gold syntax shows that there is still potential for syntax to help SRL.

| | CoNLL-05 | | CoNLL-2012 Dev. | | | | | | |
|---|---|---|---|---|---|---|---|---|---|
| | Dev. | Test | BC | BN | NW | MZ | PT | TC | WB |
| L8+PoE | 82.7 | 84.6 | 81.4 | 82.8 | 82.8 | 80.4 | 93.6 | 84.8 | 81.0 |
| +AutoSyntax | 83.2 | 84.8 | 81.5 | 82.8 | 83.2 | 80.6 | 93.7 | 84.9 | 81.1 |

Table 7: F1 on CoNLL 2005, and the development set of CoNLL 2012, broken down by genres. Syntax-constrained decoding (+AutoSyntax) shows bigger improvement on in-domain data (CoNLL 05 and CoNLL 2012 NW).

provide experimental results in Table 7 for both CoNLL 2005 and CoNLL 2012, which consists of 8 different genres. The penalties are tuned on the two development sets separately (10000 on CoNLL 2005 and 20 on CoNLL 2012). On the CoNLL 2005 development set, the predicted syntax gives a 0.5 F1 improvement over our best model, while on in-domain test and out-of-domain development sets, the improvement is much smaller. As expected, on CoNLL 2012, syntax improves most on the newswire (NW) domain. These improvements suggest that while decoding with hard constraints is beneficial, joint training or multi-task learning could be even more effective by leveraging full, labeled syntactic structures.

## 5 Related Work

Traditional approaches to semantic role labeling have used syntactic parsers to identify constituents and model long-range dependencies, and enforced global consistency using integer linear programming (Punyakanok et al., 2008) or dynamic programs (Täckström et al., 2015). More recently, neural methods have been employed on top of syntactic features (FitzGerald et al., 2015; Roth and Lapata, 2016). Our experiments show that off-

the-shelf neural methods have a remarkable ability to learn long-range dependencies, syntactic constituency structure, and global constraints without coding task-specific mechanisms for doing so.

An alternative line of work has attempted to reduce the dependency on syntactic input for semantic role labeling models. Collobert et al. (2011) first introduced an end-to-end neural-based approach with sequence-level training and uses a convolutional neural network to model the context window. However, their best system fell short of traditional feature-based systems. Neural methods have also been used as classifiers in transition-based SRL systems (Henderson et al., 2013; Swayamdipta et al., 2016). Most recently, several successful LSTM-based architectures have achieved state-of-the-art results in English span-based SRL (Zhou and Xu, 2015), Chinese SRL (Wang et al., 2015) and dependency-based SRL (Marcheggiani et al., 2017), with little to no syntactic input. Our techniques push results to more than 3 F1 over the best syntax-based models. However, we also show that there is potential for syntax to further improve performance.

## 6 Conclusion and Future Work

We presented a new deep learning model for span-based semantic role labeling with a 10% relative error reduction over the previous state of the art. Our ensemble of 8 layer BiLSTMs incorporated some of the recent best practices such as orthonormal initialization, RNN-dropout, and highway connections, and we have shown that they are crucial for getting good results with deep models.

Extensive error analysis sheds light on the strengths and limitations of our deep SRL model, with detailed comparison against shallower models and two strong non-neural systems. While our deep model is better at recovering long-distance predicate-argument relations, we still observe structural inconsistencies, which can be alleviated by constrained A$^*$ decoding.

Finally, we posed the question of whether deep SRL still needs syntactic supervision. Despite recent success without syntactic input, we found that our best neural model can still benefit from accurate syntactic parser output via straightforward constrained decoding. Our oracle experiment, where the SRL system receives a 3 F1 gain from gold syntax showed potential for syntax to further improve deep SRL models.

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
