# Peer review of "Deep Semantic Role Labeling: What Works and What’s Next"

_ACL 2017 — decision unknown_

[Official Review · Reviewer 1 · rating 4 · confidence 5]
soundness 3 · originality 4 · clarity 4 · impact 3 · substance 4 · appropriateness 5 · meaningful comparison 5 · presentation format Oral Presentation

- General Discussion:

This paper extends Zhou and Xu's ACL 2015 approach to semantic role labeling
based on deep BiLSTMs. In addition to applying recent best practice techniques,
leading to further quantitative improvements, the authors provide an insightful
qualitative analysis of their results. The paper is well written and has a
clear structure. The authors provide a comprehensive overview of related work
and compare results to a representative set of other SRL models that hace been
applied on the same data sets.

I found the paper to be interesting and convincing. It is a welcome research
contribution that not only shows that NNs work well, but also analyzes merits
and shortcomings of an end-to-end learning approach.

- Strengths:

Strong model, insightful discussion/error analysis.

- Weaknesses:

Little to no insights regarding the SRL task itself.

[Official Review · Reviewer 2 · rating 4 · confidence 5]
soundness 3 · originality 4 · clarity 5 · impact 3 · substance 5 · appropriateness 5 · meaningful comparison 5 · presentation format Oral Presentation

This paper presents a new state-of-the-art deep learning model for semantic
role labeling (SRL) that is a natural extension of the previous
state-of-the-art system (Zhou and Xu, 2015) with recent best practices for
initialization and regularization in the deep learning literature.
The model gives a 10% relative error reduction which is a big gain on this
task. The paper also gives in-depth empirical analyses to reveal the strengths
and the remaining issues, that give a quite valuable information to the
researchers in this field. 

Even though I understand that the improvement of 3 point in F1 measure is a
quite meaningful result from the engineering point of view, I think the main
contribution of the paper is on the extensive analysis in the experiment
section and a further in-depth investigation on analysis section. The detailed
analyses shown in Section 4 are performed in a quite reasonable way and give
both comparable results in SRL literature and novel information such as
relation between accuracies in syntactic parsing and SRL. This type of analysis
had often been omitted in recent papers. However, it is definitely important
for further improvement.

The paper is well-written and well-structured. 
I really enjoyed the paper and would like to see it accepted.